# Pseudo-colouring an ECG enables lay people to detect QT-interval prolongation regardless of heart rate

**Alaa Alahmadi**[1], **Alan Davies**[2], **Markel Vigo**[1], **Caroline Jay**[1]*

**1** Department of Computer Science, The University of Manchester, Manchester, United Kingdom, **2** Division of Informatics, Imaging and Data Sciences, School of Health Sciences, The University of Manchester, Manchester, United Kingdom

* caroline.jay@manchester.ac.uk

**Data Availability Statement:** The data is held in a public repository, GitHub repository: https://github. com/mbchxaa6/Pseudo-colouring_ECGs_LQTS_

## Abstract

Drug-induced long QT syndrome (diLQTS), characterized by a prolongation of the QT-interval on the electrocardiogram (ECG), is a serious adverse drug reaction that can cause the life-threatening arrhythmia Torsade de Points (TdP). Self-monitoring for diLQTS could therefore save lives, but detecting it on the ECG is difficult, particularly at high and low heart rates. In this paper, we evaluate whether using a pseudo-colouring visualisation technique and changing the coordinate system (Cartesian vs. Polar) can support lay people in identifying QT-prolongation at varying heart rates. Four visualisation techniques were evaluated using a counterbalanced repeated measures design including Cartesian no-colouring, Cartesian pseudo-colouring, Polar no-colouring and Polar pseudo-colouring. We used a multi-reader, multi-case (MRMC) receiver operating characteristic (ROC) study design within a psychophysical paradigm, along with eye-tracking technology. Forty-three lay participants read forty ECGs (TdP risk $n = 20$, no risk $n = 20$), classifying each QT-interval as normal/ abnormal, and rating their confidence on a 6-point scale. The results show that introducing pseudo-colouring to the ECG significantly increased accurate detection of QT-interval prolongation regardless of heart rate, T-wave morphology and coordinate system. Pseudo-colour also helped to reduce reaction times and increased satisfaction when reading the ECGs. Eye movement analysis indicated that pseudo-colour helped to focus visual attention on the areas of the ECG crucial to detecting QT-prolongation. The study indicates that pseudo-colouring enables lay people to visually identify drug-induced QT-prolongation regardless of heart rate, with implications for the more rapid identification and management of diLQTS.

## Introduction

### Background and significance

An electrocardiogram (ECG or EKG) is a graphical representation of the electrical activity of the heart, widely used as a clinical tool for monitoring heart function and detecting cardiac

QT-nomogram Ref: https://doi.org/10.5281/zenodo.3940377.

**Funding:** This work was supported by the first author's (Alaa Alahmadi) sponsor (Taibah University, Kingdom of Saudi Arabia, College of Computer Science and Engineering, Yanbu) and was funded for her PhD research by Saudi Arabian Cultural Bureau (grant number TAU388), https://uksacb.org/language/en/.

**Competing interests:** The authors have declared that no competing interests exist.

pathologies [1]. An important measurement on the ECG is the QT-interval, which represents the duration of the ventricular depolarization and repolarization cycle. It is measured from the beginning of the QRS complex (illustrating ventricular depolarization) to the end of the T-wave (showing subsequent repolarization) [1, 2], as shown in Fig 1.

The QT-interval is of considerable clinical importance, primarily because its prolongation can increase the risk of a life-threatening arrhythmia known as *Torsades de Pointes* (TdP), a form of polymorphic ventricular tachycardia that is the leading cause of sudden cardiac death in young, otherwise healthy people [3–6]. Prolongation of the QT-interval indicates a cardiac disorder known as 'long QT syndrome'(LQTS), which is caused by the malfunction of cardiac ion channels impairing ventricular repolarization; it manifests as a longer QT-interval than normal on the ECG [7, 8]. The TdP arrhythmia is often precipitated by triggers such as physical activity (notably swimming) [9], and stress-related emotion [10]. Athletes with LQTS are thus particularly at risk of TdP episodes [9, 11, 12]. A major difficulty with identifying LQTS is that it is often asymptomatic; sudden cardiac death can be the first clinical manifestation, and therefore it may go undiagnosed, or underdiagnosed, without ECG assessment [13–15].

LQTS can be congenital—a result of genetic mutations in cardiac channelopathies, as seen in Romano–Ward syndrome [16]—or acquired, resulting from the clinical administration of certain drugs [3, 17]. There are many commonly prescribed medications that can prolong the QT-interval, including antihistamines, antibiotics, antidepressants and antiarrhythmic drugs [3, 17]. Drug interactions that block the human ether-a-go-go-related gene (hERG) potassium channel are the most common cause of drug-induced long QT syndrome (diLQTS) [18, 19]. These drugs are associated with large and increasing numbers of sudden deaths, and preventive strategies could therefore save many lives [3, 20, 21].

Frequent ECG monitoring is strongly recommended for people at risk of acquiring diLQTS, including patients on a known QT-prolonging drug and patients participating in a clinical trial testing a new drug [22, 23]. A baseline ECG, before taking a QT-prolonging drug, and follow-up ECGs to monitor the QT-interval are advisable [24, 25]. Recent innovations in home healthcare technologies have made it possible to record high-quality, clinically reliable ECG data outside of the clinical environment [26–29], but interpreting such data correctly in a timely manner still remains a major challenge [30, 31]. One approach involves monitoring patients' ECGs remotely, where the ECG report is referred to a clinician who specialises in ECG interpretation [26–29]. However, a single dose of a QT-prolonging drug could dramatically prolong the QT-interval within 24 hours for some patients, with the risk of TdP

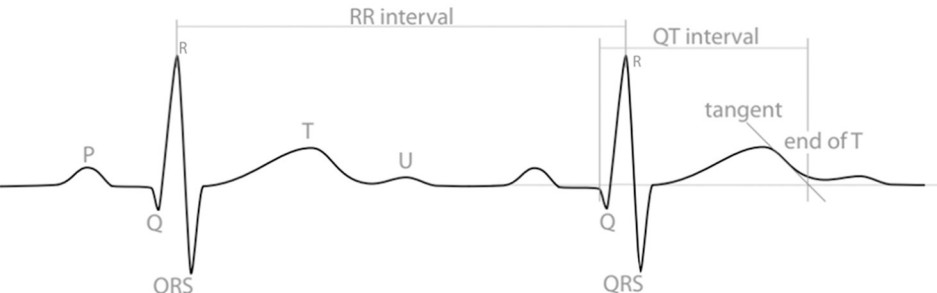

**Fig 1. An illustration of electrocardiogram (ECG or EKG) waveforms and the QT-interval.** Different 'waves' (peaks and troughs) are labelled with letters and represent different stages of a heartbeat. The QT-interval is measured from the beginning of the Q-wave to the end of the T-wave, identified here using the tangent method drawn at the maximum down-slope of the T-wave.

increasing with continued use [32]. At present, remote expert monitoring is too infrequent and costly to be an effective way of managing this issue for everyone at risk of acquiring diLQTS.

While there are computerised methods for interpreting the ECG, the accuracy of these methods remains limited [30, 31]. Challenges of automating ECG interpretation include the correct recognition of the ECG waveforms, in particular the amplitude and duration characteristics (which differ substantially across individuals), and the precise determination of the onset and offset of the different waves and complexes (P-wave, QRS complex, T-wave) [30].

Automated QT measurement algorithms have proved unsatisfactory for detecting LQTS in particular [33–39]. Garg and Lehmann [33] found that even a widely used computerized ECG analysis system was not able to detect QT-interval prolongation in 52.5% of patients affected. Research has also shown that drug-induced QT-prolongation can be underestimated and under-reported by computerised methods in patients on Methadone, a drug that is infamous for prolonging the QT-interval and increasing the risk of TdP [38]. A major challenge for automated QT algorithms is identifying the precise end of the T-wave (the terminal point), especially when the T-wave's morphology is abnormal [2, 40, 41]. This is particularly problematic, as QT-prolonging drugs also often affect the morphology of the T-wave; for example, patients on pure hERG-blocking drugs can develop flat, asymmetric, and notched T waves, whilst patients on multi-channel blocking drugs (e.g. hERG-blocking with lesser calcium and late sodium block) can develop more distorted, bizarre, T-waves [42]. In fact, specific T-wave patterns can aid detection of LQTS [43], and large T-U waves are known to precede TdP [44]. These indicators are lost by automated QT algorithms, however, as they work by abstracting the ECG data into a number (the calculated length of the QT-interval). As such, the human visual validation of QT-prolongation on the ECG remains mandatory in clinical practice, and provides the richest information for recognizing LQTS [30, 33–39].

Giving laypeople the means to detect when the QT-interval is prolonged without having to rely on an external interpreter could lead to a step-change in the detection and management of diLQTS, as well as helping to reduce the development of life-threatening complications in situations that place individuals at risk of QT-interval prolongation. ECG interpretation is complex, however, and detecting QT-prolongation on the ECG is particularly difficult, even for clinicians [45, 46]. From a perceptual-cognitive perspective, this may be related to the fact that people are poor at perceiving quantity represented along a horizontal scale [47–49]. Research has also shown that changes in the T-wave morphology and/or artifacts in the ECG signal can cause misinterpretation of the QT-interval length [40, 41, 50, 51]. The effect of heart rate on the QT-interval is another challenge, as it is the proportionate rather than absolute length that is important, and it is common to misinterpret the QT-interval at heart rates that differ from the 'standard' 60 bpm [52].

Visualisation techniques have the potential to highlight abnormal changes within the ECG, by supporting intuitive visual perception of the issues. Pre-attentive processing theory, which outlines a set of visual properties known to be detected rapidly and accurately by the human eye, is considered to be especially powerful for designing effective visualisations [53–55]. Colour is one of the most effective pre-attentive attributes that is noticed without conscious effort [56, 57]. A useful technique is pseudo-colouring, which encodes continuously varying values using a sequence of colours [55]. It has been widely used to support diagnosis from medical images, including breast disease [58], and for highlighting details in organs and bones structures that would otherwise be difficult to perceive [59, 61]. It is also used extensively in geographic and time series visualisations, where applications include encoding elevation in the data or showing changes over time [55, 60]. To our knowledge, no prior work has used pseudo-colouring to support ECG interpretation.

In this paper, we investigate whether a pseudo-colouring technique can reliably show prolongation of the QT-interval on an ECG, such that it be identified by a lay person. In an earlier feasibility study, we found that superimposing pseudo-colouring on the ECG using a spectrum-approximation colour sequence significantly improved people's ability to detect increases in the QT-interval at a low normal heart rate, when compared with a reference ECG stimulus showing a normal QT baseline [51]. This initial investigation had several limitations that affected the generalisability of the results. Firstly, the ECGs all had a single, regular heart rate of 60 bpm, and it is unknown how the technique would perform at higher or lower heart rates. The ECGs belonged to the same patient, who was taking the QT-prolonging drug ('Dofetilide', an antiarrhythmic drug with a pure hERG blocker) and subsequently experienced a gradual increase in the QT-interval from the normal QT baseline. It is well-known that the ECG differs from one individual to another, and that different types of drug can affect the ECG in different ways. The study examined only sensitivity (identifying true positives); for self-monitoring, it is important also to examine specificity (identifying true negatives). Finally, and most importantly, the study design allowed people to compare the ECG with a normal baseline. Such a baseline would be hard to produce in self-monitoring situations, not least because it would have to be adjusted according to heart rate. Therefore, this study tests the technique by presenting the ECGs one by one, asking participants to judge whether QT-prolongation has occurred when viewing just a single image, rather than comparing it with a baseline.

## Objective

The objective of the study was to evaluate the effectiveness of the pseudo-colouring technique in displaying QT-prolongation risk threshold on the ECG, when the data is displayed on two coordinate systems: Cartesian and Polar. Accuracy was quantified in terms of both sensitivity and specificity. The technique was tested with QT-intervals across several heart rates for multiple patients on different QT-prolonging drugs. No comparison baseline ECG was provided.

## Materials and methods

In this section, we provide a detailed description of how the ECGs images were produced and displayed, and the experiment designed to evaluate the effectiveness of the technique. The ECGs shown in the figures are reduced in size for inclusion in the paper. All of the full size ECG images used in the experiment, along with the R scripts used to create them, can be found in [62].

### ECG data acquisition

The ECG datasets were acquired from a clinical trial study that was conducted to assess the effects of different types of known QT-prolonging drugs, including a pure hERG potassium channel blocker and multichannel blockers, on the ECG of healthy subjects [63]. The open ECG datasets are available online from the PhysioNet database [64].

### Visualisation design

The visualisation was created in two layers. The first layer plotted the ECG using the standard method, which is a two dimensional line graph displaying the amplitude, or voltage, of the electrical signal along the Y-axis and the time in milliseconds (ms) along the X-axis. We used the ECG data from lead II, as the QT-interval is conventionally measured in this lead [2, 17]. The pseudo-colouring was then applied in a second layer, following a series of steps:

identifying individual heartbeats via R-peak detection; applying pseudo-colouring to each heartbeat; adjusting the colouring according according to heart rate; displaying the data on either a Cartesian (current standard) or Polar coordinate system.

**Identifying individual heartbeats.** The R-wave peaks (points with greatest amplitude) were detected in the raw ECG datasets using an automated numerical math function that finds the greatest peaks (maxima) in a time series [65]. A solid vertical line was superimposed on the image at this point perpendicular to the X-axis to show the location of the R-waves on the ECG, and delineate individual heartbeats. Detecting the R wave in the vast majority of ECGs is straightforward, as it consistently has the greatest amplitude. This is in contrast to the other waveforms which vary considerably across ECGs. As high quality ECG signals were used in the study, this method proved accurate and efficient. Where signal data is lower quality, it may be necessary to use alternative detection methods that can reliably distinguish R-peaks from high amplitude noises (see e.g. [66–69]).

**Applying the pseudo-colouring.** The traditional way to measure the QT-interval clinically is to count the small squares (each representing 40ms) on the standard ECG background grid, from the beginning of the Q-wave to the end of the T-wave. For the purposes of applying the pseudo-colouring, we added sequential time in ms as a third dimension to each heartbeat, from the R-peak minus 20ms (the estimated start of the Q-wave), to the clinically significant point of maximum risk for TdP—the method for estimating this is provided in the next section. We then applied pseudo-colouring to the area under the curve of the ECG signal by mapping every 40ms of this sequential time to a colour code or hue, with the intensity of the hue changing every millisecond. We used a spectrum-approximation sequence as it is effective for supporting people in reading continuous values [70]; it ranges from cool colours (purple to blue to green), which were used to indicate normal QT-interval levels, to warm colours (yellow to orange to red), used to show prolonged QT-interval levels.

**Adjusting pseudo-colouring according to heart rate using the QT nomogram.** To apply the pseudo-colouring accurately for different heart rates, it was necessary to identify the value of QT-prolongation that determines being 'at risk' for TdP. In the previous study [51], where all ECGs had a 60 bpm heart rate, we used the half R-R interval rule to identify at risk QT-prolongation. This states that the QT-interval is prolonged if it is equal to or longer than the half R-R interval (midpoint between two consecutive R-peaks) [52, 71]. This rule is accurate with a heart rate of 60 bpm, but lacks sensitivity in detecting at risk QT-prolongation with both lower and higher heart rates [52, 72]. In clinical practice it is common to apply a QT correction formula (QTc), and then use a 'cut off' value to identify at risk QT-prolongation. Examples of such QTc formulae include Bazett's, Fredericia's, Framingham's and Hodge's [52]. However, recent research has shown that these correction formulae are inaccurate in identifying patients at risk of drug-induced TdP, particularly for fast and slow heart rates [52, 73].

An alternative approach, known as the 'QT nomogram', is a risk assessment method designed specifically for identifying patients at risk of drug-induced TdP according to heart rate [74, 75]. It was developed after screening 129 ECG cases that reported drug-induced TdP and comparing these with control cases (i.e. when no TdP was reported). The actual QT-interval value (not the QTc) and the heart rate for each case were plotted as coordinates on a graph; this produced a line showing the upper bound of the QT-interval value at risk for TdP as a function of heart rate. If the QT/HR value falls on or above this risk line, the patient is at risk of TdP; below the line the patient is not considered at risk of TdP [74]. The nomogram plot can be seen in Fig 2. Further evaluation of the QT nomogram showed that it had higher sensitivity and specificity than widely accepted QTc formulas [76] and the half R-R interval rule

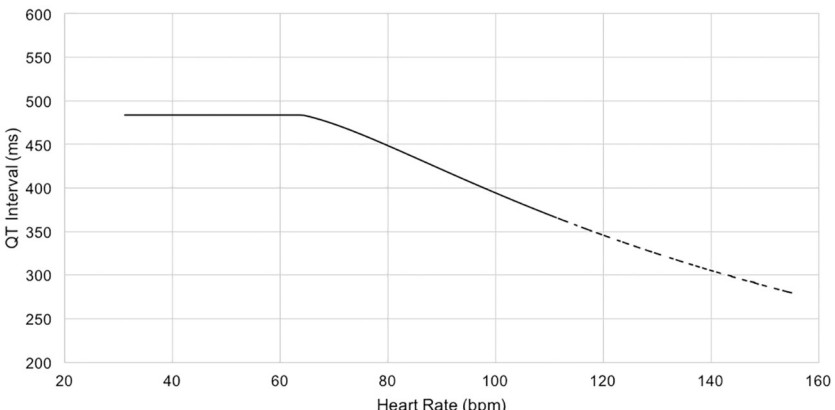

**Fig 2. The QT-nomogram for identifying QT-prolongation at risk of TdP over heart rate.** If the QT/HR value falls on or above the risk line, the patient is at risk of TdP; below the line the patient is not considered at risk of TdP [74].

[72]. We therefore used it as the foundation for modifying the pseudo-colour as a function of heart rate.

For each ECG, the heart rate (HR) was calculated and then the corresponding QT value for the 'at risk' threshold was acquired from the QT nomogram and plotted on the ECG as a dashed vertical line. To apply the pseudo-colouring, we used nine indices, where each was mapped to a colour code and represented by a small square on the standard ECG background grid. A time value on the nomogram threshold line was mapped to dark orange, and values 40ms and 80ms above the nomogram line were mapped to red and dark red respectively, showing a higher risk for TdP. Time values below the nomogram line were mapped to progressively cooler colours, such that a time value below the line by five small squares (200ms), was mapped to blue. Fig 3 illustrates how the pseudo-colouring technique was applied according to the standard ECG background grid. The mappings between colours and time values according

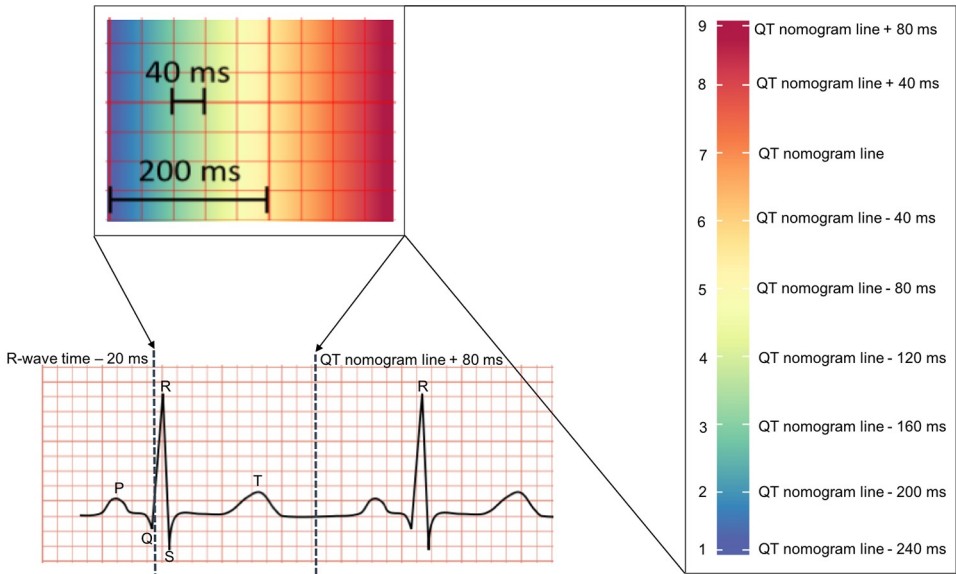

**Fig 3. Mapping the pseudo-colouring to the ECG, according to the QT-nomogram.** A small square on the standard ECG background grid is equal to 40ms.

Table 1. The nine indices on the pseudo-colouring scale with their corresponding time values and colour codes.

| Index | The corresponding time value (ms) | Colour code |
|:-:|---|---|
| 1 | QT-value of the nomogram line—$(40 \times 6)$ | Purple |
| 2 | QT-value of the nomogram line—$(40 \times 5)$ | Blue |
| 3 | QT-value of the nomogram line—$(40 \times 4)$ | Green |
| 4 | QT-value of the nomogram line—$(40 \times 3)$ | Lime |
| 5 | QT-value of the nomogram line—$(40 \times 2)$ | Yellow |
| 6 | QT-value of the nomogram line—$(40 \times 1)$ | Orange |
| 7 | QT-value of the nomogram line | Dark orange |
| 8 | QT-value of the nomogram line + $(40 \times 1)$ | Red |
| 9 | QT-value of the nomogram line + $(40 \times 2)$ | Dark red |

to the nomogram is shown in Table 1. Note that this approach does not provide a numerical value for the QT-interval, but rather colours the area under the curve of the signal over a specific time interval (40ms × 9 indices = 360ms) according to the nomogram. The T-wave area lies within this time interval: if the T-wave is in a cool colour location, then the QT-interval is in the normal range; if it is in a warm colour location, the interval is prolonged.

**Coordinate system.** Our previous work indicates that people may be able to detect QT-prolongation more accurately on a Polar coordinate system [51]. To see whether this held when ECGs with varying heart rates are presented in isolation, we compared two coordinate systems (Cartesian and Polar) with and without pseudo-colouring. We used the R programming language [77] using RStudio software version 1.1.447 to create the visualisations. Fig 4 shows examples of ECGs with pseudo-colouring that have different heart rates, but similar QT-interval risk levels for TdP, on Cartesian coordinates. Fig 5 shows the same ECGs on Polar coordinates.

## Experimental design

We used a multi-reader, multi-case (MRMC) receiver operating characteristic (ROC) study design within a psychophysical paradigm. The MRMC ROC design is an evaluation method commonly used to assess diagnostic performance in medical imaging studies, where multiple human observers (readers) interpret multiple patient images (cases) [78]. The area under the ROC curve, a plot of sensitivity versus 1-specificity, is used to measure diagnostic accuracy. The MRMC design helps to increase the generalisability of study results and enhance statistical power, particularly when evaluating different computer-assisted detection (CAD) systems [78]. Psychophysical experiments use detection and discrimination tasks to investigate the relationship between the intensity of a physical stimulus and human perception and sensation, by systematically varying the properties of the stimulus along one or more physical dimensions [79]. Here the paradigm was used to systematically evaluate the impact of the visualisation technique on people's sensitivity to increases in the QT-interval. We also used eye-tracking, a non-invasive sensor technology that measures human eye movements, to help us interpret the visual behaviour underpinning the results on a *post hoc* basis [80].

The study used a counterbalanced repeated measures design with two independent variables, each with two levels:

1. Colour-coding (no colouring and pseudo-colouring).

2. Coordinate system (Cartesian and Polar).

The within-subjects factorial design yielded a total of 4 (2×2) experimental conditions for each participant: Cartesian no-colouring; Cartesian pseudo-colouring; Polar no-colouring;

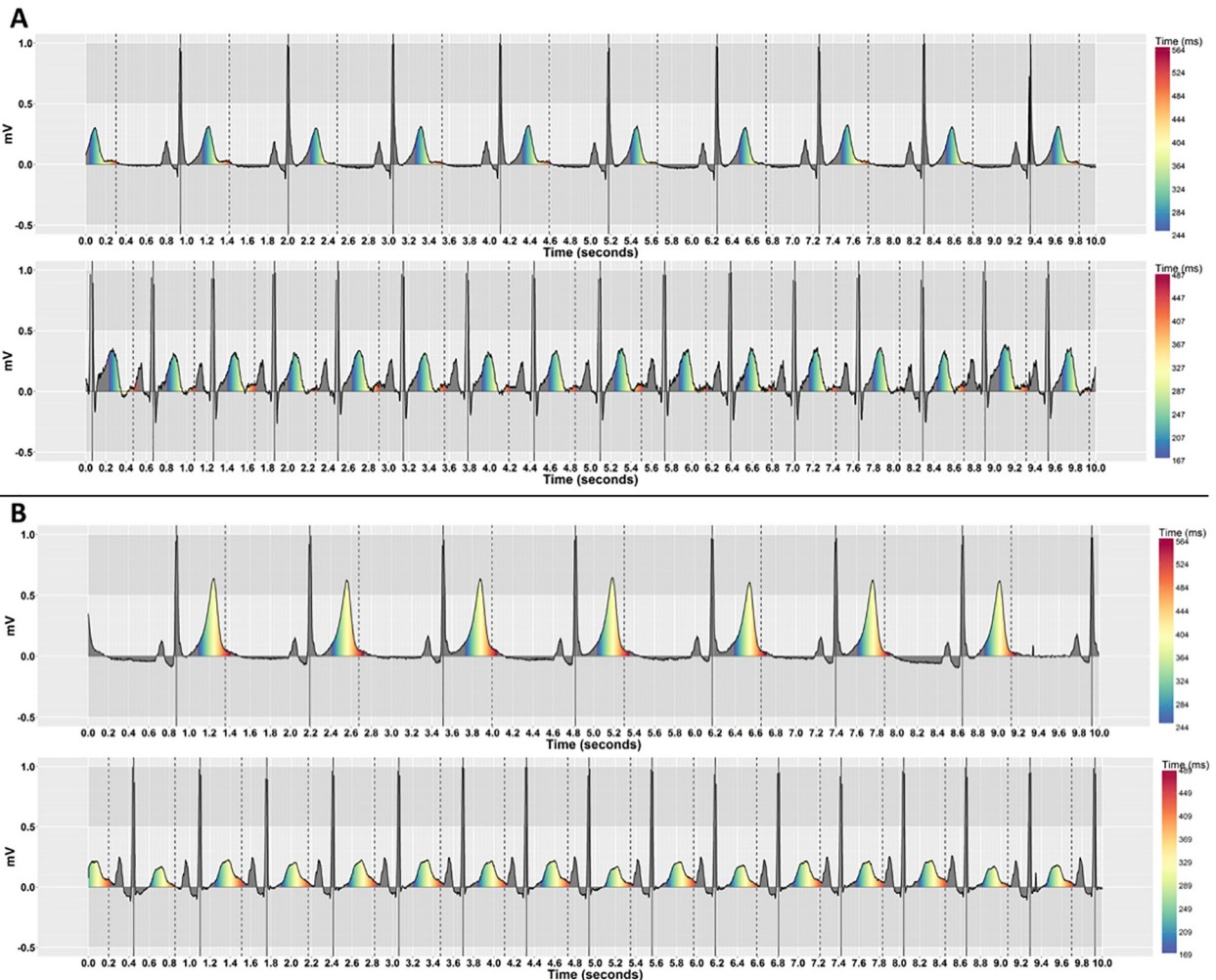

**Fig 4. Pseudo-coloured ECGs that have different heart rates, but similar QT-interval risk levels, on Cartesian coordinates.** (A) QT-intervals are below the nomogram line by 80ms for both ECGs (no risk of TdP). The top stimulus has a low heart rate (HR = 57, QT = 401) and the bottom has a high heart rate (HR = 95, QT = 339). (B) QT-intervals are on the nomogram line in both ECGs (risk of TdP). The top stimulus has a low heart rate (HR = 46, QT = 487) and the bottom has a high heart rate (HR = 94, QT = 410).

Polar pseudo-colouring. We counterbalanced the order of visualisation presentation using a balanced Latin square to minimize practice effects. The dependent variables were accuracy (broken down into sensitivity and specificity), reaction time in ms, satisfaction with the visualisation on a five point Likert-type scale from low (1) to high (5), mean fixation duration, total fixation duration and spatial fixation distribution.

## Stimulus design

The heart rates of the ECGs acquired from the clinical trial ranged from 40 to 96 bpm, and the QT-interval values ranged from 300 to 579ms. As such, bradycardia (HR <60 bpm) was included in the range of heart rates, but not tachycardia (HR >100 bpm).

We followed psychophysical experimental design principles to systematically select the study's stimuli. We considered QT interval value, relative to the nomogram line, to be the physical dimension along which the stimulus was varied, in units of 40ms. Six levels of the QT-interval were identified: three levels below the nomogram line (no/low risk to borderline), one

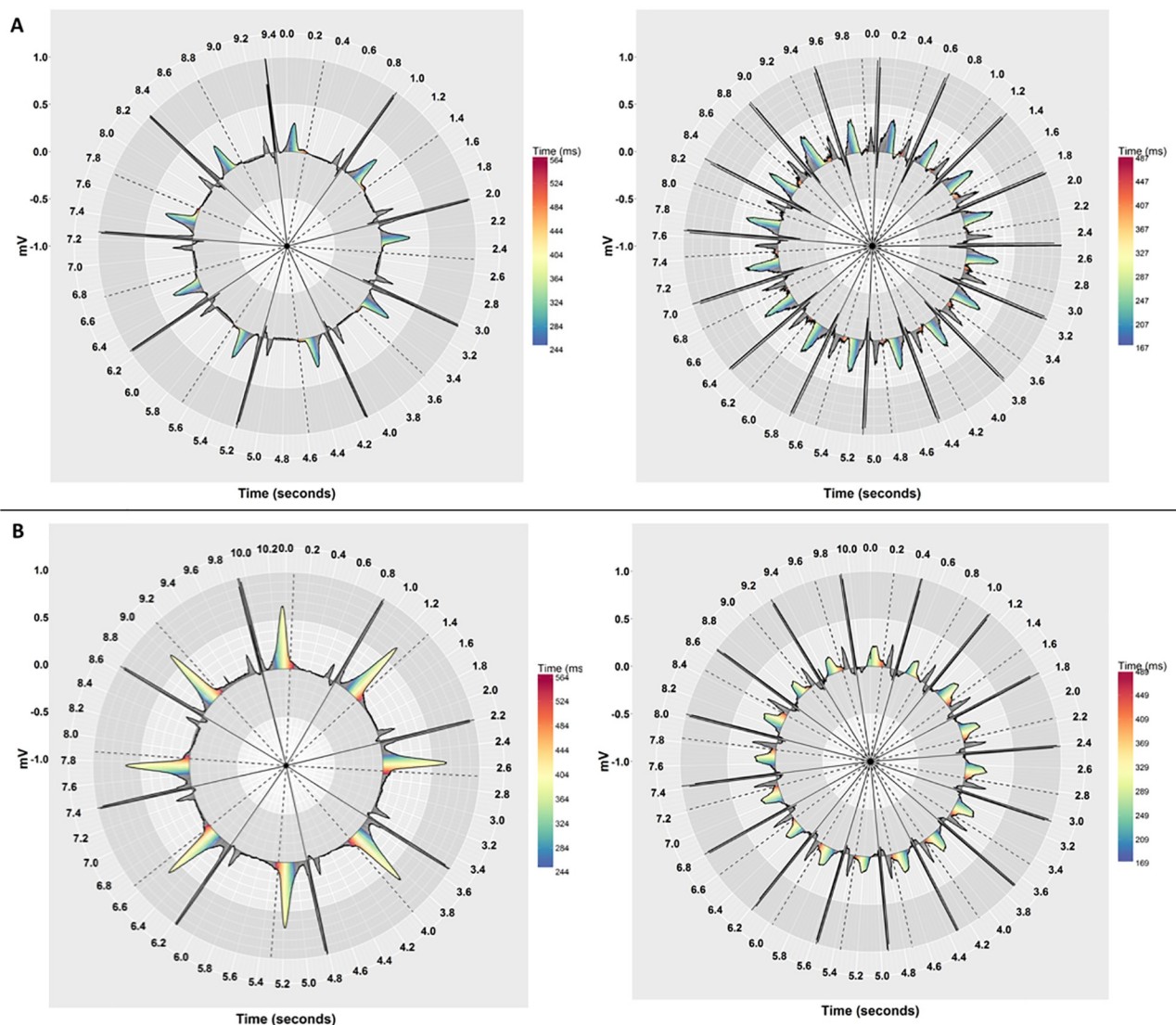

**Fig 5. Pseudo-coloured ECGs that have different heart rates, but similar QT-interval risk levels, on Polar coordinates.** (A) QT-intervals are below the nomogram line by 80ms for both ECGs (no risk of TdP). The left stimulus has a low heart rate (HR = 57, QT = 401) and the right has a high heart rate (HR = 95, QT = 339). (B) QT-intervals are on the nomogram line in both ECGs (risk of TdP). The left stimulus has a low heart rate (HR = 46, QT = 487) and the right has a high heart rate (HR = 94, QT = 410).

level on the nomogram line (moderate risk), and two levels above the nomogram line (high risk). The six QT levels were represented by indices 4 to 9 on the pseudo-colouring scale respectively (see Table 1). Fig 6 shows examples of ECGs with the first QT-level (normal), and the sixth QT-level (severely prolonged), while Figs 4 and 5 show examples of ECGs with the second QT-level (normal) and the fourth QT-level (prolonged). Table 2 shows the corresponding estimated QT-interval prolongation for the six levels.

## ECG case selection

Forty ECG cases were selected from multiple patients ($n$ = 17) to match the six QT-levels. Twenty ECGs were below the nomogram line (i.e. no risk of TdP) and 20 ECGs were on or above the nomogram line (i.e. at risk of TdP). Six ECG cases were on a placebo, and had values

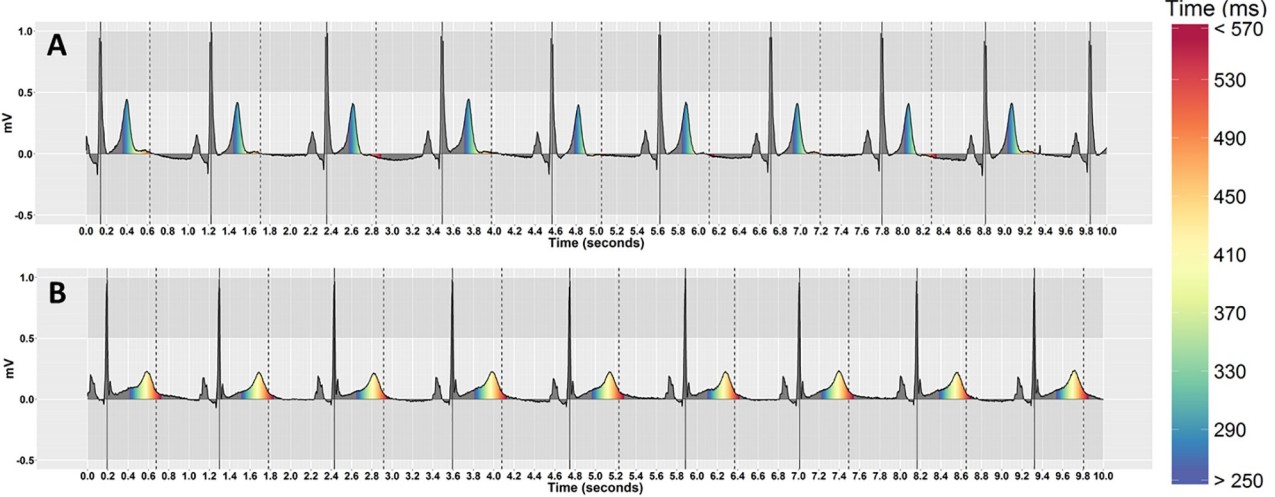

**Fig 6. Pseudo-coloured ECGs that have similar heart rates, but different QT-interval risk levels, on Cartesian coordinates.** (A) Normal QT-interval (HR = 55, QT = 361). (B) Severely prolonged QT-interval (HR = 52, QT = 579).

corresponding to the first QT-level, across heart rates ranging from 48 to 90 bpm. The other 34 ECG cases (18 on Dofetilide, a known QT-prolonging drug with a pure hERG potassium channel blocker, and 16 on Quinidine, a multichannel blocker with a strong hERG block, and lesser calcium and late sodium blocks), represented the second to sixth QT-levels across heart rates from 42 to 96 bpm. Table 3 shows the metadata of each ECG case in terms of QT value, heart rate, drug type, patient ID, the corresponding QT-level relative to the nomogram line and whether the case was at risk for TdP or not based on the nomogram plot.

## Participants

A total of forty three participants, with no experience in ECG interpretation, were recruited from a university campus (30 students and 13 staff). Eligibility for the study was determined by asking participants to rate their knowledge of ECG interpretation, and only people who identified as having no knowledge at all were included in the study. There were 18 male and 25 female participants between 20 and 56 years old (Mean = 31, *SD* = 8). Table 4 shows the participants' demographic data. Participants' sight was normal or corrected-to-normal and they reported no motor or neurological disorders. The research was approved by the University Research Ethics Committee). All participants provided written informed consent.

## Apparatus

Stimuli were displayed on a 23.8 inch (diagonal) wide-screen Spectrum eye-tracking monitor, with a resolution of 1920 × 1080 pixels. Eye gaze was recorded using the Tobii Pro Spectrum

**Table 2. The six QT-levels with their corresponding QT-value relative to the nomogram and an estimated range.**

| QT-level | QT-value relative to the QT-nomogram line | Estimated range |
|---|---|---|
| 1 | QT-value of the nomogram line—(40 × 3) | Normal |
| 2 | QT-value of the nomogram line—(40 × 2) | Normal |
| 3 | QT-value of the nomogram line—(40 × 1) | Borderline |
| 4 | QT-value of the nomogram line | Prolonged |
| 5 | QT-value of the nomogram line + (40 × 1) | Very prolonged |
| 6 | QT-value of the nomogram line + (40 × 2) | Severely prolonged |

**Table 3. Characteristics of the 40 selected ECG cases used in the study.**

| ECG ID | QT value | Heart rate | Drug type | Patient ID | QT-level | Risk for TdP on the nomogram |
|--------|----------|-----------|-----------|-----------|----------|------------------------------|
| 1 | 370 | 48 | Placebo | 11 | 1 | Not at risk |
| 2 | 361 | 55 | Placebo | 13 | 1 | Not at risk |
| 3 | 350 | 68 | Placebo | 16 | 1 | Not at risk |
| 4 | 343 | 72 | Placebo | 16 | 1 | Not at risk |
| 5 | 329 | 83 | Placebo | 18 | 1 | Not at risk |
| 6 | 335 | 90 | Placebo | 4 | 1 | Not at risk |
| 7 | 401 | 57 | Dofetilide | 12 | 2 | Not at risk |
| 8 | 389 | 75 | Dofetilide | 18 | 2 | Not at risk |
| 9 | 339 | 95 | Dofetilide | 20 | 2 | Not at risk |
| 10 | 419 | 47 | Quinidine | 8 | 2 | Not at risk |
| 11 | 396 | 68 | Quinidine | 16 | 2 | Not at risk |
| 12 | 355 | 82 | Quinidine | 18 | 2 | Not at risk |
| 13 | 445 | 46 | Dofetilide | 2 | 3 | Not at risk |
| 14 | 441 | 67 | Dofetilide | 7 | 3 | Not at risk |
| 15 | 431 | 75 | Dofetilide | 8 | 3 | Not at risk |
| 16 | 417 | 80 | Dofetilide | 11 | 3 | Not at risk |
| 17 | 371 | 94 | Dofetilide | 17 | 3 | Not at risk |
| 18 | 444 | 58 | Quinidine | 4 | 3 | Not at risk |
| 19 | 424 | 76 | Quinidine | 10 | 3 | Not at risk |
| 20 | 363 | 95 | Quinidine | 22 | 3 | Not at risk |
| 21 | 487 | 46 | Dofetilide | 4 | 4 | At risk |
| 22 | 468 | 72 | Dofetilide | 8 | 4 | At risk |
| 23 | 451 | 79 | Dofetilide | 9 | 4 | At risk |
| 24 | 445 | 81 | Dofetilide | 10 | 4 | At risk |
| 25 | 486 | 54 | Quinidine | 7 | 4 | At risk |
| 26 | 485 | 64 | Quinidine | 7 | 4 | At risk |
| 27 | 419 | 91 | Quinidine | 15 | 4 | At risk |
| 28 | 410 | 94 | Quinidine | 19 | 4 | At risk |
| 29 | 523 | 42 | Dofetilide | 4 | 5 | At risk |
| 30 | 494 | 71 | Dofetilide | 9 | 5 | At risk |
| 31 | 470 | 85 | Dofetilide | 18 | 5 | At risk |
| 32 | 565 | 49 | Dofetilide | 4 | 6 | At risk |
| 33 | 579 | 52 | Dofetilide | 18 | 6 | At risk |
| 34 | 547 | 64 | Dofetilide | 21 | 6 | At risk |
| 35 | 518 | 54 | Quinidine | 4 | 5 | At risk |
| 36 | 509 | 68 | Quinidine | 7 | 5 | At risk |
| 37 | 482 | 80 | Quinidine | 9 | 5 | At risk |
| 38 | 417 | 96 | Quinidine | 21 | 5 | At risk |
| 39 | 518 | 77 | Quinidine | 21 | 6 | At risk |
| 40 | 507 | 79 | Quinidine | 21 | 6 | At risk |

The metadata were acquired from the clinical trial study [63], published by PhysioNet database [64].

eye-tracker and Tobii Pro Lab 1.95 software with a sampling rate of 600Hz. Key press events were recorded during the experiment to measure reaction times. Each Cartesian coordinate ECG stimulus measured 32.31cm × 6.14cm, and each Polar coordinate stimulus was 15.61cm × 12.93cm.

**Table 4. Participant demographics.**

| Category | Count | Percentage % |
|---|---:|---:|
| Age | | |
| 20-30 | 25 | 58% |
| 31-40 | 13 | 30% |
| 41-50 | 3 | 7% |
| 50+ | 2 | 5% |
| Sex | | |
| Male | 18 | 42% |
| Female | 25 | 58% |
| Occupation | | |
| University Staff | 13 | 30% |
| Student | 30 | 70% |
| Background | | |
| Computer science | 27 | 63% |
| Chemical engineering | 1 | 2% |
| Psychology | 3 | 7% |
| Biochemistry | 4 | 9% |
| Interior design | 1 | 2% |
| Mathematics | 1 | 2% |
| Art, history and sociology | 1 | 2% |
| Biomedical sciences | 4 | 9% |
| Business | 1 | 2% |

## Task and procedure

Participants completed a 15 minute training session prior to starting the experiment, where they were introduced to the ECG trace, and taught how to identify the QT-interval using the tangent method to identify the end of the T-wave (see Fig 1 and [81]). Participants were instructed to use the tangent method visually on the screen (i.e. without any additional equipment) to roughly identify the end of the T-wave; the training session did not involve any medical terms or high-level training techniques typically associated with clinical ECG interpretation and focused solely on identification of the QT interval. Participants were then introduced to the four visualisation techniques, and shown how to use the pseudo-colouring scale and the risk threshold (represented by a dashed line) to assess the QT-interval. Participants were told that a QT-interval is considered prolonged if the T-wave ends on or exceeds the risk threshold dashed line, and when the pseudo-colouring within the QT-interval, and particularly in the Tpeak–Tend interval, contains warm colours (yellow/orange/red), with a greater visibility of warm colours indicating a higher level of QT-prolongation. Finally, participants completed an assessment task, in which they were asked to highlight the start and end points of the QT-intervals on two different ECGs and assess whether it is normal or abnormal using the visualisation techniques, to ensure that they understood how to perform the task. Participants were tested individually. They completed four tests, one for each visualisation technique, in which they read all ECG cases for one test before moving to the next. Participants read the same ECG cases using each visualisation technique. ECGs were presented at random within a test. Each test began with a five-point calibration of the eye-tracking apparatus. ECGs were presented one at a time, and participants completed two tasks: the first task used a psychophysical one alternative forced-choice paradigm (also known as a yes-no detection task), in which

participants indicated verbally, as quickly as possible, whether they perceived the QT-interval as ('normal' or 'abnormal'), whilst pressing the space-bar on the keyboard to collect reaction time; the second task was to rate their confidence in their response using a 6-point scale of 'very likely normal' (1), 'probably normal' (2), 'possibly normal' (3), 'possibly abnormal' (4), 'probably abnormal' (5), and 'very likely abnormal' (6). There was no time limit imposed, but participants were encouraged to do the tasks as quickly as possible. The binary responses and the confidence scores were recorded on a spreadsheet during the experiment by the researcher. At the end of the experiment, participants were asked to provide their satisfaction with the visualisations by rating how much each helped them to assess the QT-interval using a five point Likert-type scale ranging from 'not very much' (1) to 'a lot' (5).

## Statistical analysis methods

Response accuracy was measured by calculating the average area under the receiver operating characteristic (ROC) curves (AUC) with 95% confidence intervals (CIs). QT-levels 1-3 inclusive were classified as negative (normal), and levels 4-6 as positive (prolonged). We used the Dorfman-Berbaum-Metz multi-reader multi-case (DBM MRMC) software from the University of Iowa [82], to calculate and compare the AUCs of the four visualisation techniques, based on the methods of Dorfman, Berbaum and Metz [83] and Obuchowski and Rockette [84] and later unified and improved by Hillis and colleagues [78, 85, 86]. The DBM MRMC method uses jackknifing and analysis of variance (ANOVA) methods and we considered both readers and cases as random variables; this allows the results to be generalised to the population of readers and cases. The ROC curve was fitted using the trapezoidal area and Wilcoxon method. The statistical power of ROC MRMC was greater than 90%, and the sample size estimation was performed using [87].

All statistical tests were performed at a 5% significance level ($\alpha$ = 0.05); two-sided 95% confidence intervals (CIs) were used to quantify uncertainty. Within-participant comparisons of sensitivity and specificity were performed using McNemar's chi-squared test. We measured perceptual sensitivity to QT-levels using the psychometric function and just noticeable difference (JND) threshold, along with the signal detection analysis method R packages [88, 89]. Differences in reaction time and satisfaction scores were determined via a Friedman test and *post hoc* pairwise comparisons performed with a Wilcoxon signed-rank test utilising a Bonferroni correction ($\alpha$ = 0.008). Visual behaviour was analysed using eye tracking metrics including mean fixation duration, total fixation duration and fixation distribution using the nearest neighbour index (NNI) method, as implemented by Davies *et al.* [90].

## Results

All anonymised data and related metadata underpinning the findings reported in this article can be found in [62].

### Accuracy

**Area under the ROC curve.** Pseudo-colouring significantly increased the average area under the ROC curve (AUC) for both coordinate systems (Tables 5 and 6). When Cartesian coordinates were used, the average AUC increased from 0.895 (standard error (SE) = 0.024) to 0.935 (SE = 0.020), while the increase was from 0.878 (SE = 0.029) to 0.934 (SE = 0.023) for Polar coordinates (Table 5). The average increase in AUC as a result of pseudo-colouring was therefore 0.04 for Cartesian and 0.056 for Polar coordinates, a statistically significant increase in both cases ($p$ = 0.014, $p < 0.001$ for Cartesian and Polar coordinates respectively, see Table 6). There was no significant difference in average AUC as a function of coordinate

**Table 5. The average area under the ROC curve across all participants and for each visualisation technique.**

| Visualisation technique | Average AUC | SE | %95 CI |
|---|---|---|---|
| Cartesian no-colouring | 0.895 | 0.024 | 0.848 to 0.943 |
| Cartesian pseudo-colouring | 0.935 | 0.020 | 0.895 to 0.974 |
| Polar no-colouring | 0.878 | 0.029 | 0.821 to 0.935 |
| Polar pseudo-colouring | 0.934 | 0.023 | 0.888 to 0.980 |

SE = Standard error. CI = %95 Confidence intervals.

system when pseudo-colouring was used ($p = 0.978$, see Table 6). Although the average AUC was higher for Cartesian coordinates than for Polar coordinates when no-colouring was used, the difference was not statistically significant ($p = 0.273$, see Table 6).

**Specificity.** Specificity was calculated as the proportion of true negative (i.e. 'normal') responses for QT-levels 1 to 3. The mean specificity across all participants was high and similar for both coordinate systems regardless of pseudo-colouring was used (Table 7). Although pseudo-colouring increased the mean specificity for Polar coordinates from 0.88 (SE = 0.002) to 0.90 (SE = 0.002), the increase was not statistically significant ($p = 0.460$). The highest mean specificity was 0.94 (SE = 0.002) for Cartesian coordinates when pseudo-colouring was not used, which was significantly higher than when pseudo-colouring was used with this coordinate system ($p < 0.001$). There was no difference in the mean specificity between the two coordinate systems when pseudo-colouring was used ($p = 1.000$).

**Table 6. Pairwise comparisons of the area under the ROC curve for the four visualisation techniques.**

| Pairwise comparisons | AUC Difference | SE | P-value | 95% CI |
|---|---|---|---|---|
| Cartesian no-colouring, Cartesian pseudo-colouring | -0.039 | 0.015 | **0.014** | -0.070 to -0.007 |
| Cartesian no-colouring, Polar no-colouring | 0.017 | 0.015 | 0.273 | -0.013 to 0.048 |
| Cartesian no-colouring, Polar pseudo-colouring | -0.038 | 0.015 | **0.015** | -0.070 to -0.007 |
| Cartesian pseudo-colouring, Polar no-colouring | 0.056 | 0.015 | **<0.001** | 0.025 to 0.087 |
| Cartesian pseudo-colouring, Polar pseudo-colouring | 0.000 | 0.015 | 0.978 | -0.030 to 0.031 |
| Polar no-colouring, Polar pseudo-colouring | -0.056 | 0.015 | **<0.001** | -0.087 to -0.024 |

Significant p-values are in bold. SE = Standard error. CI = %95 Confidence intervals.

**Table 7. Comparisons of mean specificity between the visualisation techniques.**

| Coordinate | Pseudo-colouring | No-colouring | Difference | P-value |
|---|---|---|---|---|
| Cartesian | 0.90±0.002, | 0.94±0.002, | −0.04±0.002, | **<0.001** |
|  | 0.870 to 0.929 | 0.913 to 0.966 | -0.069 to -0.014 |  |
| Polar | 0.90±0.002, | 0.88±0.002, | 0.02±0.002, | 0.460 |
|  | 0.874 to 0.925 | 0.849 to 0.910 | -0.024 to 0.044 |  |
| Difference | 0.00±0.002, | 0.06±0.002, |  |  |
|  | -0.030 to 0.030 | 0.0260 to 0.073 |  |  |
| P-value | 1.000 | **<0.001** |  |  |

The table shows the mean ± standard error, and 95% two-sided confidence intervals. Significant p-values are in bold and were calculated using a McNemar's chi-squared test.

**Table 8. Comparisons of mean sensitivity between the visualisation techniques.**

| Coordinate | Pseudo-colouring | No-colouring | Difference | P-value |
|---|---|---|---|---|
| Cartesian | 0.83±0.003, | 0.63±0.004, | 0.2±0.004, | **<0.001** |
|  | 0.773 to 0.877 | 0.570 to 0.689 | 0.138 to 0.261 |  |
| Polar | 0.82±0.003, | 0.72±0.004, | 0.1±0.003, | **<0.001** |
|  | 0.779 to 0.860 | 0.665 to 0.774 | 0.050 to 0.149 |  |
| Difference | 0.01±0.003, | −0.09±0.003, |  |  |
|  | -0.030 to 0.050 | -0.140 to -0.039 |  |  |
| P-value | **0.001** | **<0.001** |  |  |

The table shows the mean ± standard error, and 95% two-sided confidence intervals. Significant p-values are in bold and were calculated using a McNemar's chi-squared test.

**Sensitivity.** Sensitivity was measured using the ROC method as the proportion of true positive (i.e. 'abnormal') responses for QT-levels 4 to 6. The mean sensitivity increased with pseudo-colouring from 0.63 (SE = 0.004) to 0.83 (SE = 0.003) with Cartesian coordinates, and from 0.72 (SE = 0.004) to 0.82 (SE = 0.003) with Polar coordinates (Table 8). The increase in sensitivity of 0.2 (SE = 0.004) with Cartesian and 0.1 (SE = 0.003) with Polar coordinates was statistically significant ($p < 0.001$). Sensitivity was higher when Cartesian coordinates were used than when Polar coordinates were used by 0.01, a statistically significant difference ($p = 0.001$). When pseudo-colouring was not used, the mean sensitivity was 0.09 (SE = 0.003) higher with Polar coordinates than with Cartesian coordinates, a difference that was statistically significant ($p < 0.001$).

We also measured perceptual sensitivity across QT-levels using the following psychophysical detection methods:

*(1) Psychometric function and just noticeable difference (JND).* We used a psychometric function, an inferential model used in psychophysical detection and discrimination tasks, to model the relationship between QT-level increases and participants' sensitivity across the four visualisation techniques. The psychometric function was plotted as the proportion of responses that indicated a QT-prolongation as a function of the QT-level. The results show that pseudo-colour significantly improved the perception of QT-interval increases regardless of heart rate with both coordinate systems (see Fig 7A). We estimated the just noticeable difference (JND) threshold, which is defined in psychophysics as the smallest change required in a dimension of a stimulus in order for a difference to be perceived [79]. In this study, we defined it as the minimum increase in the QT-interval with respect to the nomogram line required for it to be detectable. We estimated the 75% JND threshold as the value of the QT-interval increase from the nomogram line at which the proportion of responses indicating QT-prolongation is equal to 0.75. The JND thresholds, determined by fitting the psychometric function using a logistic function with maximum likelihood estimation (MLE) (Fig 7B), were 51.2, 6.9, 28.8 and 7.3 milliseconds for *Cartesian no-colouring*, *Cartesian pseudo-colouring*, *Polar no-colouring* and *Polar pseudo-colouring* respectively. Pseudo-colour thus reduces the JND in both coordinate systems, showing that 75% of people are able to perceive a QT-interval as prolonged when the QT-value above the nomogram line increases by approximately 6.9 and 7.3ms with Cartesian and Polar coordinates respectively. When pseudo-colouring was not used, the JND was 51.2 and 28.8ms for Cartesian and Polar coordinates respectively, showing that sensitivity was higher for Polar than Cartesian coordinates.

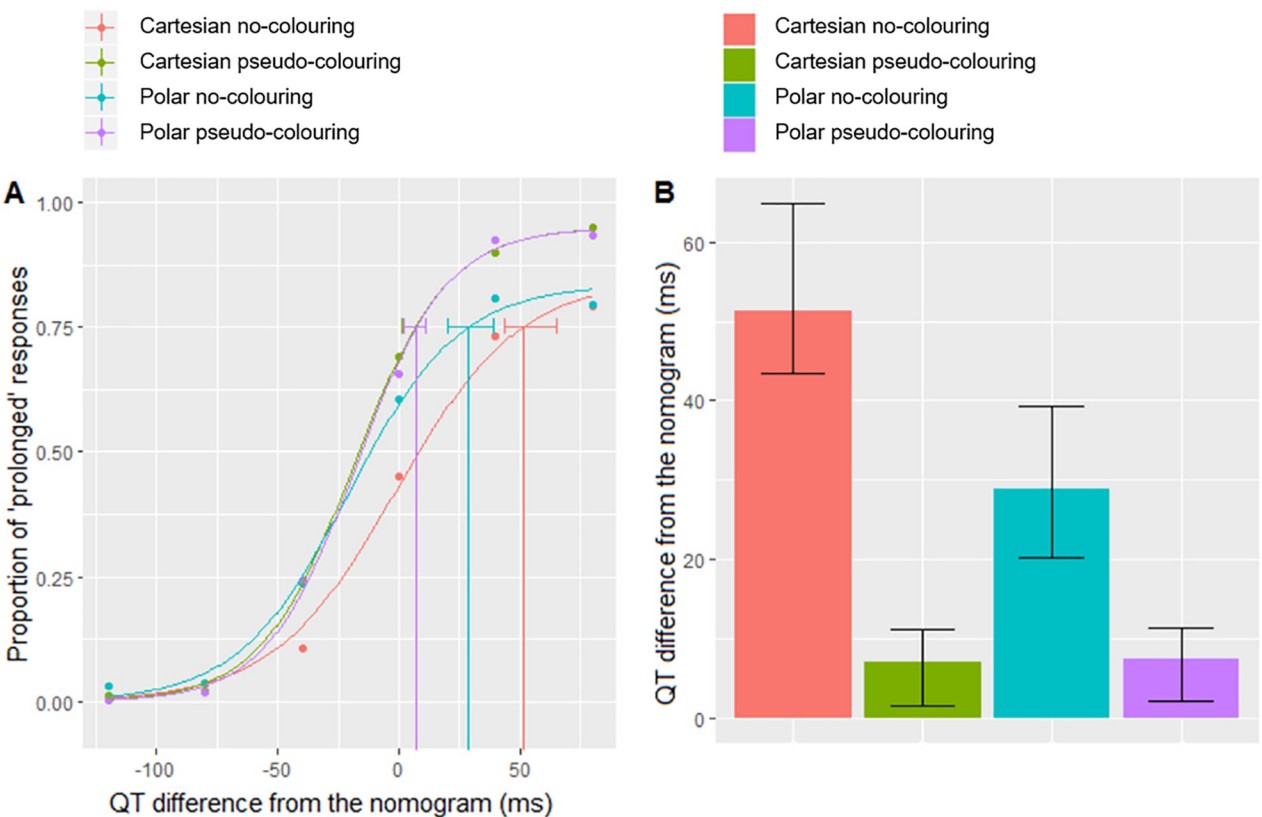

**Fig 7. Psychophysical detection measures of sensitivity.** (A) The fitted psychometric function plot shows the proportion of responses indicating QT-prolongation as a function of the QT-interval difference from the nomogram line, which corresponds to the six QT-levels. The QT value of the nomogram line (i.e. the fourth QT-level) is equal to 0 on the X-axis. Cartesian no-colouring is represented by a red line, Cartesian pseudo-colouring by a green line, Polar no-colouring by a turquoise line and Polar pseudo-colouring by a purple line. (B) The just noticeable difference (JND) thresholds plot. The error bars represent bootstrap confidence intervals.

*(2) Signal detection analysis.* The psychometric function and JND threshold quantify sensitivity in terms of a binary response (here, 'normal' or 'abnormal'). In psychophysics, signal detection theory recognises that there is uncertainty in the task, such that the decision may be affected by noise either externally, for example where a stimulus changes between presentations, or internally, as people may be biased towards saying 'normal' or 'abnormal' more frequently [79]. We attempted to minimise the effect of decision bias by increasing sample size, counterbalancing experimental conditions, and including confidence scores, but some uncertainty is likely to remain. Signal detection theory measures the sensitivity, while taking response bias into account, using a sensitivity index $d'$ [79]. This was calculated for each participant and then averaged across all participants. Table 9 shows the mean sensitivity index ($d'$) with the standard deviation and standard error for each visualisation technique. The results show that pseudo-colouring increased the mean sensitivity with both coordinate systems independent of decision bias.

## Reaction time

We measured reaction time as the period from the appearance of the stimulus on the screen to the key press event when people made their binary decision. The mean reaction times across all stimuli were 7.9 and 8.6 seconds for Cartesian coordinates with and without pseudo-

**Table 9. The mean sensitivity index for each visualisation technique.**

| Visualisation technique | Sensitivity index ($d'$) | SD | SE |
|---|---|---|---|
| Cartesian no-colouring | 1.87 | 0.55 | 0.01 |
| Cartesian pseudo-colouring | 2.22 | 0.49 | 0.01 |
| Polar no-colouring | 1.82 | 0.57 | 0.01 |
| Polar pseudo-colouring | 2.15 | 0.44 | 0.01 |

SD = Standard deviation. SE = Standard error.

**Table 10. Results of the Friedman test comparing reaction times across the four visualisation techniques.**

| QT-level | Estimated Range | Risk for TdP | $\chi^2(3)$ | p-value |
|---|---|---|---|---|
| 1 | Normal | No risk | 35.251 | **<0.001** |
| 2 | Normal | No risk | 26.307 | **<0.001** |
| 3 | Borderline | No risk | 16.349 | **0.001** |
| 4 | Prolonged | At risk | 18.216 | **<0.001** |
| 5 | Very Prolonged | At risk | 32.223 | **<0.001** |
| 6 | Severely prolonged | At risk | 48.008 | **<0.001** |
| Overall | All ECG stimuli | | 141.737 | **<0.001** |

colouring respectively, and 8.1 and 8.9 seconds for Polar coordinates with and without pseudo-colouring respectively. A Friedman test was conducted for each QT-level stimulus and overall for each condition. People were significantly faster when pseudo-colouring was used for each QT-level and over all ECG stimuli cumulatively ($p \leq 0.001$)(Table 10, Fig 8). A *post hoc* pairwise comparison using a Wilcoxon signed-rank test with Bonferroni correction ($\alpha = 0.008$) showed that although participants responded faster when pseudo-colouring was used

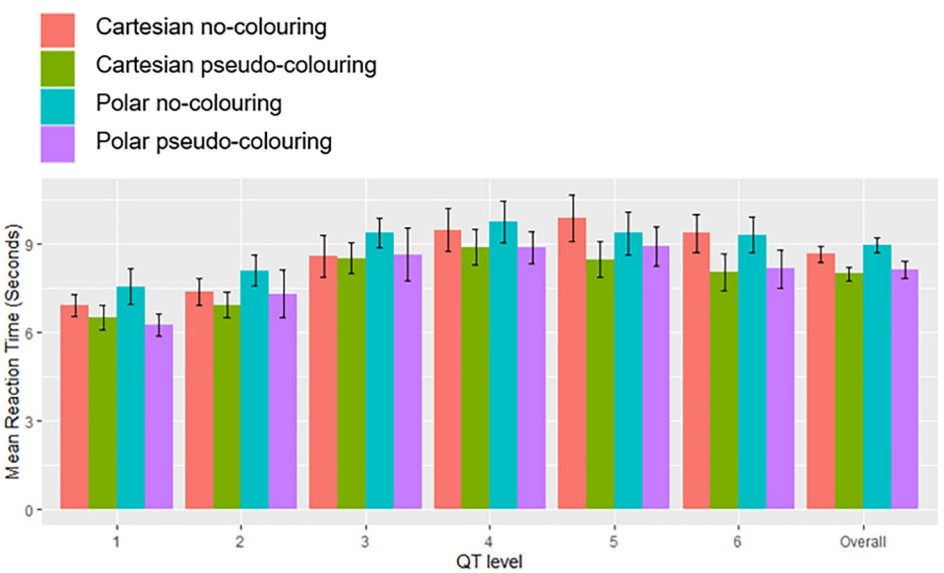

**Fig 8. Mean reaction time in seconds for each QT-level and all stimuli combined.** Error bars represent 95% confidence intervals.

with Cartesian coordinates than with Polar coordinates, the difference was not significant ($p = 0.808$).

## Satisfaction

A Friedman test showed there was a statistically significant difference in participants' satisfaction as a function of visualisation technique ($\chi^2(3) = 50.954$, $p < 0.001$). A *post hoc* pairwise comparison using a Wilcoxon signed-rank test with Bonferroni correction ($\alpha = 0.008$) showed that people preferred pseudo-colouring to no-colouring ($p < 0.008$); and Cartesian coordinates to Polar coordinates whether pseudo-colouring was used ($Z = -3.340^c$, $p = 0.001$), or not ($Z = -3.029^c$, $p = 0.002$). Although people were more sensitive to QT-prolongation when the ECG was presented on Polar coordinates with pseudo-colouring, people preferred Cartesian coordinates without pseudo-colouring to Polar coordinates with pseudo-colouring ($Z = -2.142^b$, $p = 0.032$).

## Eye movement analysis

An area of interest (AOI) was created on the whole ECG for each stimulus. As such, 160 AOIs were created (40 ECGs × 4 visualisation techniques).

**Mean fixation duration.** The mean fixation duration metric, which is an indicator of cognitive load [91, 92], was calculated across all participants for each ECG stimulus and then averaged over each QT-level. Mean fixation duration increased as the QT-level increased when pseudo-colouring was used (Fig 9A). When considered alongside the psychometric function (Fig 7A), this indicates that pseudo-colour helped people to focus visual attention. When pseudo-colouring was not used, mean fixation duration continued to increase at higher QT-levels for Cartesian coordinates, but this effect was not seen with Polar coordinates, indicating a difference in how people interpreted the ECG as a function of coordinate system.

**Total fixation duration.** The total fixation duration (also known as the dwell time) quantifies the amount of time spent fixating on the stimulus [91]. It was calculated across all

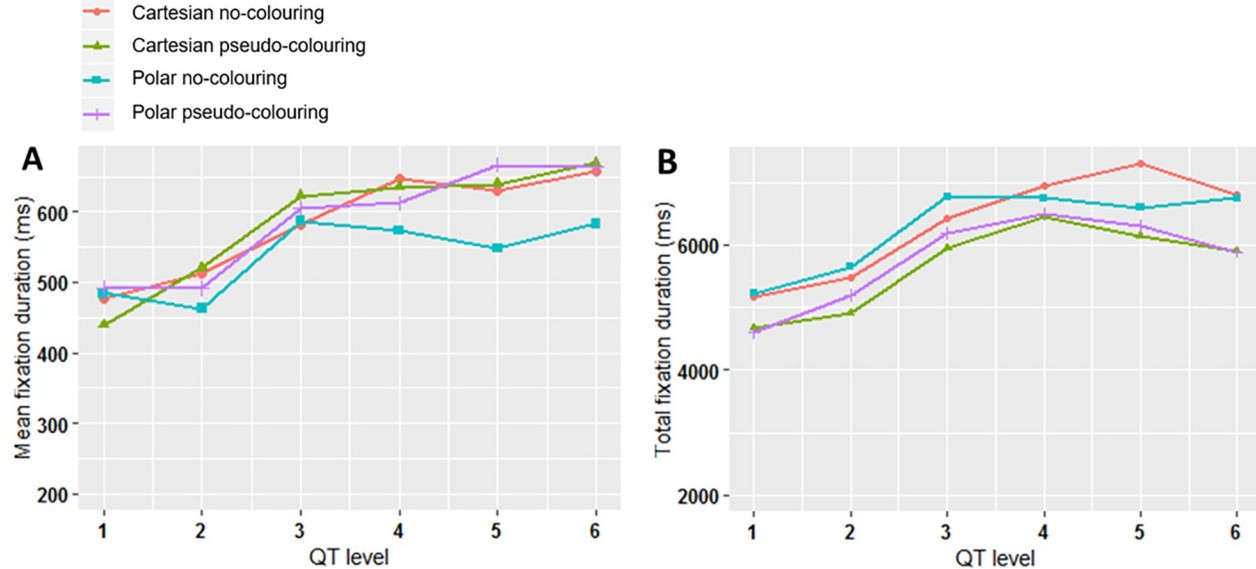

**Fig 9. Eye movement analysis results.** (A) Mean fixation duration in milliseconds on the ECG stimuli averaged for each QT-level. (B) Total fixation duration in milliseconds on the ECG stimuli averaged for each QT-level.

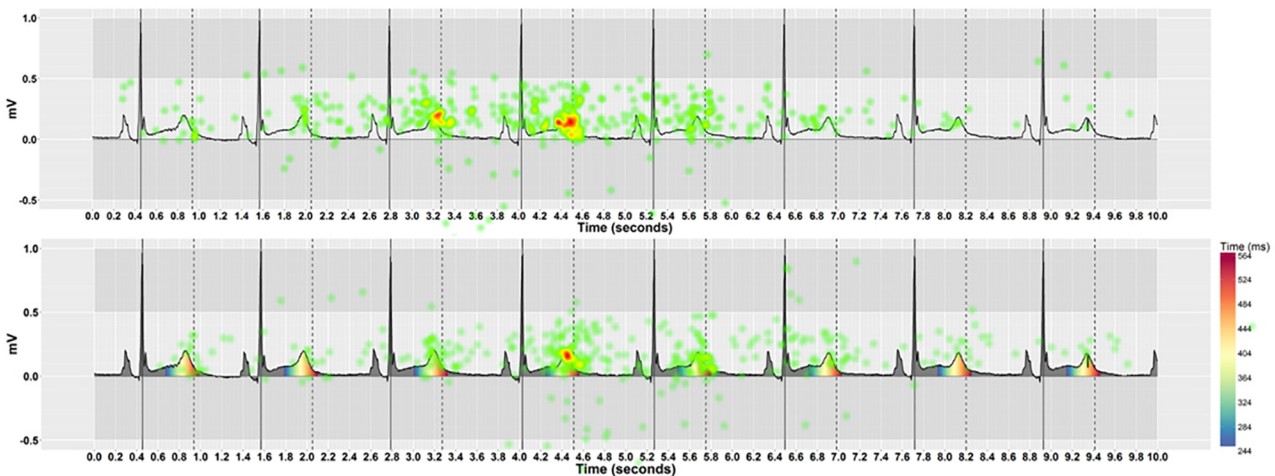

**Fig 10. A heatmap of fixation count shows the cumulative number of fixations across all participants on Cartesian coordinates.** With pseudo-colouring, people made fewer, more clustered fixations on the coloured T-wave area compared with no-colouring on the same ECG.

participants for each stimulus and then averaged for each QT-level. The results show that total fixation duration was lower when pseudo-colouring was used for both coordinate systems (Fig 9B). People spent less cumulative time fixating on the ECG stimuli with pseudo-colouring, but individual fixations were more focused (Fig 9A and 9B). When pseudo-colouring was not used, there were more, shorter fixations, indicating a less-focused interpretation strategy.

**Fixation distribution.** Figs 10 and 11 show heatmaps of fixations across an ECG stimulus with and without pseudo-colouring, for Cartesian and Polar coordinate systems respectively. We hypothesised that pseudo-colour helped to draw people's attention to the coloured T-wave area, and that fixations would therefore be more clustered when pseudo-colour was used. We

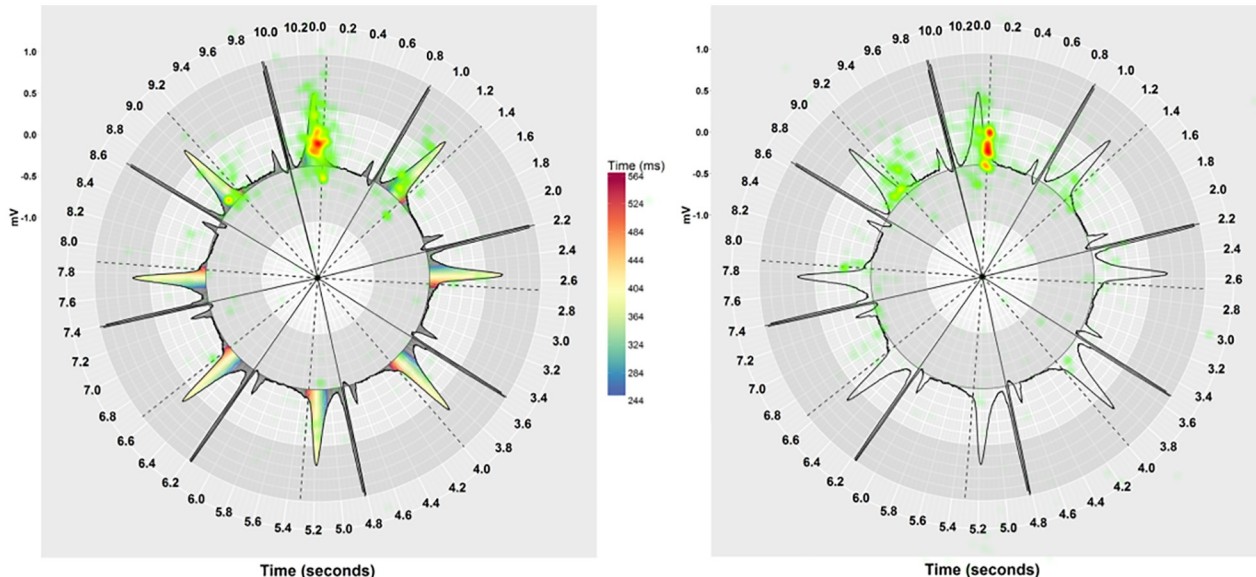

**Fig 11. A heatmap of fixation duration shows the cumulative mean fixation duration across all participants on Polar coordinates.** With pseudo-colouring, fixations were longer on the coloured T-wave area and the vertical dashed line that represents the QT risk threshold, while with no-colouring fixations were longer on the vertical dashed line only.

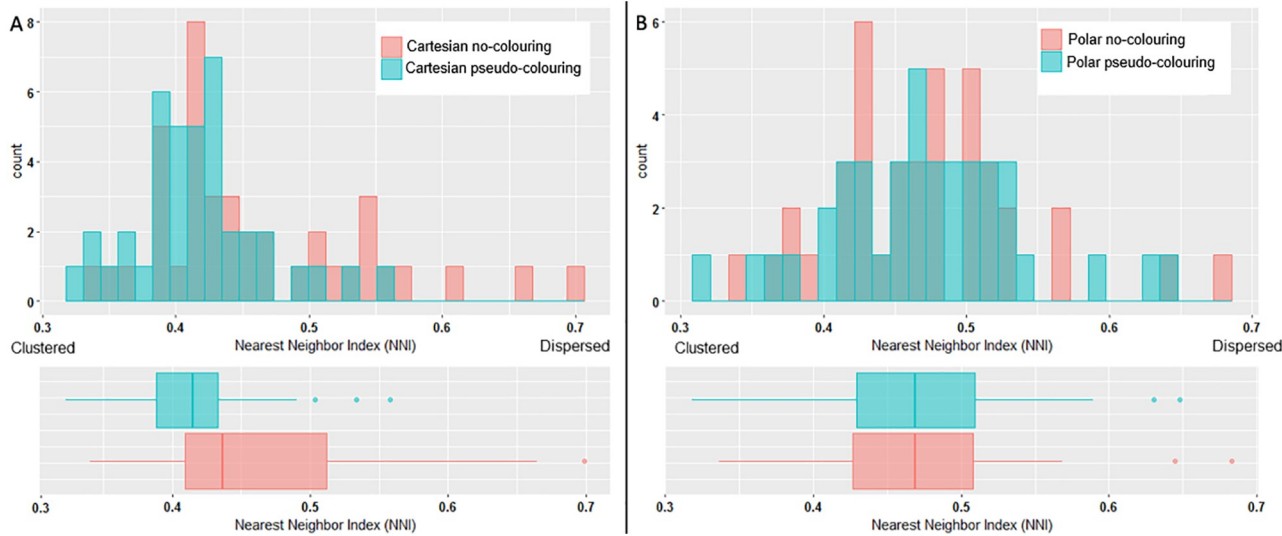

**Fig 12. Nearest Neighbour Index (NNI) of fixations on ECGs with and without pseudo-colouring.** (A) Cartesian coordinates. (B) Polar coordinates.

used the Nearest Neighbour Index (NNI) to determine if the fixations were randomly spaced or clustered. The NNI provides a ratio of the distribution pattern of points in space from 0 (clustered) to 1 (dispersed) [90]. We calculated the average Nearest Neighbour Index (NNI) for each ECG stimulus across all participants and for all visualisation techniques. Fig 12 shows a histogram of the distribution of the Nearest Neighbour Index (NNI), along with a box-plot for Cartesian and Polar coordinates, with and without pseudo-colouring. When Cartesian coordinates were used (Fig 12A), pseudo-colouring resulted in clustering in more stimuli than no-colouring. When Polar coordinates were used (Fig 12B), however, pseudo-colouring did not appear to affect the fixation distribution.

## Discussion

This study demonstrates that applying pseudo-colouring to an ECG relative to the QT nomogram significantly increases lay people's accuracy in visually assessing QT-intervals regardless of heart rate (Tables 5 and 6). In particular, pseudo-colouring increased sensitivity to prolongation (true positive cases); whilst people were able to identify normal QT-intervals without colour, adding pseudo-colouring resulted in more accurate identification of at risk QT-prolongation cases, shown by both the ROC curve analysis and the psychophysical detection measures of sensitivity (Tables 8 and 9; Fig 7). The psychometric function shows that pseudo-colour helped increase sensitivity to QT-interval increases even before a patient is at risk of TdP (Fig 7A), specifically in the borderline range (i.e. the third QT-level). This level could be critical from a clinical perspective as just a 40ms increase from this point could put someone at risk of TdP. Clinical research has shown that even small (∼10ms) QT-interval increases from the baseline should be regarded as a significant side effect of a QT-prolonging drug [93–95]. From the perspective of our study design, this meant that pseudo-colouring with Cartesian coordinates reduced specificity (true negative cases) at this level, as we considered it to be within the normal range. From a practical perspective, however, considering this level to be abnormal may serve a useful purpose.

Pseudo-colour significantly reduced reaction times and helped to focus visual attention on the areas of the ECG crucial for detecting QT-prolongation. Eye-tracking data showed that

pseudo-colour consistently increased mean fixation duration as the QT-level increased (Fig 9A); when the ECG was presented on Cartesian coordinates, fixations were more clustered when pseudo-colouring was used (Fig 12A).

Obtaining a precise measurement of the QT-interval is known to be challenging. An important QT-interval increase may be just a few milliseconds, less than one small square on the ECG. This would be particularly difficult to detect at high heart rates, as the QT-interval length shrinks with the R-R interval length [40, 41]. Here, we demonstrate that pseudo-colour can support detection of QT prolongation regardless of heart rate without needing to measure the QT-interval (Figs 4 and 5). Fig 11 shows a heat map of absolute fixation duration across all participants on an ECG with a QT-interval 40ms above the nomogram line. When pseudo-colouring was not used, people's fixations appear to be more focused on measuring the gap between the end of the T-wave and the QT risk threshold dashed line. When pseudo-colouring is present on the same ECG, fixations appear more focused on the T-wave. Focusing attention in this area is particularly useful, as long QT syndrome (LQTS) is associated with prolonging ventricular repolarization [7, 8], which is represented by the T-wave on the ECG [1, 2]. Research examining the effects of QT-prolonging drugs is investigating precisely how the T-wave responds to them to provide further insights of drug ion channel interactions and TdP risk [42]. Pseudo-colour might ultimately help clinicians to determine which part of the T-wave— the initial half of the T-wave (J-T-peak) or the second half (T-peak T-end)—underpins QT-prolongation.

## Limitations and future work

Whilst this work indicates the potential for pseudo-colouring to assist in the self-monitoring of QT-interval, the study took place in a controlled setting with a limited number of stimuli, and the transferability of the technique to practice remains an open question. Cartesian and Polar coordinates support the same level of accuracy, but people expressed a preference for pseudo-colour displayed on Cartesian coordinates. It may be, however, that the ECG trace would be better presented with Polar coordinates on smaller screens like smart watches, and future work should thus examine the effects of screen size and lighting setting on accuracy.

In this study, we only investigated the assessment of QT-interval, and it is not clear whether these visualisation techniques would generalise to interpretation of other ECG abnormalities, such as changes in ST-segment elevation. Research has shown that a large number of clinicians lack the skills to interpret QT-prolongation accurately [45, 46]; pseudo-colouring could also be used to help clinicians, especially within emergency departments, to visually assess and monitor patients' QT-intervals before or during the provision of a QT-prolonging medication. Future work should evaluate the visualisation techniques in clinical trials with more diverse clinical populations.

## Conclusion

Applying pseudo-colouring to ECGs according to the QT-nomogram supports lay people in detecting QT-prolongation visually, regardless of heart rate. The results indicate that self-monitoring ECGs for drug-induced LQTS is feasible, with the potential to prevent the development of life threatening complications.

## Acknowledgments

We would like to thank the Digital Experimental Cancer Medicine Team (digitalECMT), Manchester, UK, for their feedback and support.

## Author Contributions

**Conceptualization:** Alaa Alahmadi, Caroline Jay.

**Data curation:** Alaa Alahmadi.

**Formal analysis:** Alaa Alahmadi.

**Methodology:** Alaa Alahmadi.

**Supervision:** Markel Vigo, Caroline Jay.

**Visualization:** Alaa Alahmadi.

**Writing – original draft:** Alaa Alahmadi.

**Writing – review & editing:** Alaa Alahmadi, Alan Davies, Markel Vigo, Caroline Jay.

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
