## [Decision Letter · Decision Letter 0]

4 Jun 2020

PONE-D-20-02458

Pseudo-colouring ECGs enables lay people to detect drug-induced QT-prolongation regardless of heart rate: A multi-reader, multi-case ROC, psychophysical and eye-tracking study

PLOS ONE

Dear Dr. Alahmadi,

Thank you for submitting your manuscript to PLOS ONE. After careful consideration, we feel that it has merit but does not fully meet PLOS ONE’s publication criteria as it currently stands. Therefore, we invite you to submit a revised version of the manuscript that addresses the points raised during the review process.

Please correct the manuscript according to the comments of all reviewers. Please answer all the comments of the reviewers point by point.

We look forward to receiving your revised manuscript.

Kind regards,

Paweł Pławiak, D.Sc., Ph.D.

Academic Editor

PLOS ONE

Journal Requirements:

Additional Editor Comments (if provided):

Please correct the manuscript according to the comments of all reviewers. Please answer all the comments of the reviewers point by point.

Reviewers' comments:

Reviewer's Responses to Questions

**Comments to the Author**

1. Is the manuscript technically sound, and do the data support the conclusions?

Reviewer #1: Partly

Reviewer #2: Yes

2. Has the statistical analysis been performed appropriately and rigorously? 

Reviewer #1: Yes

Reviewer #2: Yes

3. Have the authors made all data underlying the findings in their manuscript fully available?

Reviewer #1: Yes

Reviewer #2: Yes

4. Is the manuscript presented in an intelligible fashion and written in standard English?

Reviewer #1: Yes

Reviewer #2: Yes

5. Review Comments to the Author

Reviewer #1: The authors proposed a pseudo-coloring ECG method to investigate drug-induced QT-prolongation sing eye-tracking study. In this study, the manual inspections of four types of ECG visualizations techniques are explored. The authors have performed an extensive level of statistical analysis to support their results. Overall, the article is well presented. However, the authors need to clarify the following reviewer's concerns for possible publication.

1. Please improve the quality of the figures from Fig.4.

2. The authors are encouraged to provide the details of the multi-reader multi-case Roc and eye-tracking method.

3. Why authors not followed popular R-peak detection methods? Because the proposed scheme accuracy depends on the correct annotations of ECG events (P-QRS-T and QT interval).

4. In the case of true negative cases, no-coloring schemes are giving better results, whereas, in the case of true positives, the scenario becomes opposite, colored schemes are outperforming. The authors are suggested to add this justification in the discussion section.

Reviewer #2: Please change the title too lengthy and not clear. Make a crisp title.

It is just study or real contribution is there?

The motivation behind pseudo color any suh approaches in literature, needs a clear explanation.

6. PLOS authors have the option to publish the peer review history of their article (what does this mean?). If published, this will include your full peer review and any attached files.

Reviewer #1: No

Reviewer #2: No

---

## [Author Response · Author response to Decision Letter 0]

11 Jul 2020

Thanks to the reviewers for their comments and suggestions. Below we outline how we addressed each point raised by the reviewers.

Reviewer 1

1- Please improve the quality of the figures from Fig.4

We have improved the quality of all figures and double-checked them using the PACE tool provided by PLoS. We also uploaded all full size stimuli used in the study to our GitHub repository, which is referenced in the paper.

2- The authors are encouraged to provide the details of the multi-reader multi-case Roc and eye-tracking method.

We have now explained the methods in more detail in the ‘experimental design’ subsection. 

3- Why authors not followed popular R-peak detection methods? Because the proposed scheme accuracy depends on the correct annotations of ECG events (P-QRS-T and QT interval).

The pseudo-colouring visualisation technique proposed in the paper does not require any prior annotations of ECG waveforms (P-QRS-T and QT interval); it only requires R-peak detection. As the ECG signals acquired from the clinical trial study were high quality, our R-peak detection method was accurate and efficient. We agree that with low quality ECG signals, using other R-peak detection methods that can reliably distinguish R-peaks from high amplitude noises is essential. We now added this clarification and suggestion under the ‘Identifying individual heartbeats’ subsection.

4- In the case of true negative cases, no-coloring schemes are giving better results, whereas, in the case of true positives, the scenario becomes opposite, colored schemes are outperforming. The authors are suggested to add this justification in the discussion section.

We now discussed these findings in more detail in the ‘discussion’ section, showing why the specificity (true negative cases) decreased with pseudo-colouring, which was mainly at the QT-level three/borderline to the nomogram.

Reviewer 2

1- Please change the title too lengthy and not clear. Make a crisp title.

We have now shortened the title

2- It is just study or real contribution is there? 

To the best of our knowledge, no prior studies have examined how and to what extent using pseudo-colouring technique can support ECG interpretation. This statement is now added in the ‘background and significance’ section.

3- The motivation behind pseudo color any such approaches in literature, needs a clear explanation.

We have highlighted the part in the ‘background and significance’ section that explained the motivation behind the pseudo-colouring technique and provided examples of its applications in the literature.

---

## [Decision Letter · Decision Letter 1]

5 Aug 2020

Pseudo-colouring an ECG enables lay people to detect QT-interval prolongation regardless of heart rate

PONE-D-20-02458R1

Dear Dr. Alahmadi,

We’re pleased to inform you that your manuscript has been judged scientifically suitable for publication and will be formally accepted for publication once it meets all outstanding technical requirements.

Kind regards,

Paweł Pławiak, D.Sc., Ph.D.

Academic Editor

PLOS ONE

Additional Editor Comments (optional):

Reviewers' comments:

Reviewer's Responses to Questions

**Comments to the Author**

1. If the authors have adequately addressed your comments raised in a previous round of review and you feel that this manuscript is now acceptable for publication, you may indicate that here to bypass the “Comments to the Author” section, enter your conflict of interest statement in the “Confidential to Editor” section, and submit your "Accept" recommendation.

Reviewer #1: All comments have been addressed

Reviewer #2: All comments have been addressed

2. Is the manuscript technically sound, and do the data support the conclusions?

Reviewer #1: (No Response)

Reviewer #2: Partly

3. Has the statistical analysis been performed appropriately and rigorously? 

Reviewer #1: (No Response)

Reviewer #2: N/A

4. Have the authors made all data underlying the findings in their manuscript fully available?

Reviewer #1: (No Response)

Reviewer #2: No

5. Is the manuscript presented in an intelligible fashion and written in standard English?

Reviewer #1: (No Response)

Reviewer #2: No

6. Review Comments to the Author

Reviewer #1: (No Response)

Reviewer #2: please re check the presentation once. Some how not completely satisfied. But accepting it. It may be noted

7. PLOS authors have the option to publish the peer review history of their article (what does this mean?). If published, this will include your full peer review and any attached files.

Reviewer #1: No

Reviewer #2: No

---

## [Editor Report · Acceptance letter]

11 Aug 2020

PONE-D-20-02458R1 

Pseudo-colouring an ECG enables lay people to detect QT-interval prolongation regardless of heart rate 

Dear Dr. Alahmadi:

I'm pleased to inform you that your manuscript has been deemed suitable for publication in PLOS ONE. Congratulations! Your manuscript is now with our production department. 

Kind regards, 

on behalf of

Prof. Paweł Pławiak 

Academic Editor

PLOS ONE